# Development and validation of a model for individualized prediction of hospitalization risk in 4,536 patients with COVID-19

**Lara Jehi**[1]*, **Xinge Ji**[2], **Alex Milinovich**[2], **Serpil Erzurum**[3], **Amy Merlino**[4], **Steve Gordon**[5], **James B. Young**[6], **Michael W. Kattan**[2]

1 Neurological Institute, Chief Research Information Officer, Cleveland Clinic, Cleveland, Ohio, United States of America, 2 Quantitative Health Science Department, Lerner Research Institute Cleveland Clinic, Cleveland, Ohio, United States of America, 3 Respiratory Institute, Chair of the Lerner Research Institute, Cleveland Clinic, Cleveland, Ohio, United States of America, 4 Obstetrics and gynecology, Chief Medical Information Ofc., Cleveland Clinic, Cleveland, Ohio, United States of America, 5 Infectious Disease Department, Cleveland Clinic, Cleveland, Ohio, United States of America, 6 Cardiology, Chief Academic Officer, Cleveland Clinic, Cleveland, Ohio, United States of America

* jehil@ccf.org

**Data Availability Statement:** Data used for the generation of this risk prediction model includes human research participant data that are sensitive and cannot be publicly shared due to legal and

## Abstract

### Background

Coronavirus Disease 2019 is a pandemic that is straining healthcare resources, mainly hospital beds. Multiple risk factors of disease progression requiring hospitalization have been identified, but medical decision-making remains complex.

### Objective

To characterize a large cohort of patients hospitalized with COVID-19, their outcomes, develop and validate a statistical model that allows individualized prediction of future hospitalization risk for a patient newly diagnosed with COVID-19.

### Design

Retrospective cohort study of patients with COVID-19 applying a least absolute shrinkage and selection operator (LASSO) logistic regression algorithm to retain the most predictive features for hospitalization risk, followed by validation in a temporally distinct patient cohort. The final model was displayed as a nomogram and programmed into an online risk calculator.

### Setting

One healthcare system in Ohio and Florida.

### Participants

All patients infected with SARS-CoV-2 between March 8, 2020 and June 5, 2020. Those tested before May 1 were included in the development cohort, while those tested May 1 and later comprised the validation cohort.

ethical restrictions by the Cleveland clinic regulatory bodies including the institutional review Board and legal counsel. In particular, variables like the patient's address, date of testing, dates of hospitalization, date of ICU admission, and date of mortality are HIPAA protected health information and legally cannot be publicly shared. Since these variables were critical to the generation and performance of the model, a partial dataset (everything except them) is not fruitful either because it will not help in efforts of academic advancement, such as model validation or application. We will make our data sets available upon request, under appropriate data use agreements with the specific parties interested in academic collaboration. Requests for data access can be made to mascar@ccf.org.

**Funding:** None of the authors report any conflicts of interest or have any relevant disclosures. LJ, SE, and MK were funded by NIH/NCATS UL1TR002548. https://ncats.nih.gov/ The funders had no role in study design, data collection and analysis, decision to publish, or preparation of the manuscript.

**Competing interests:** The authors have declared that no competing interests exist.

## Measurements

Demographic, clinical, social influencers of health, exposure risk, medical co-morbidities, vaccination history, presenting symptoms, medications, and laboratory values were collected on all patients, and considered in our model development.

## Results

4,536 patients tested positive for SARS-CoV-2 during the study period. Of those, 958 (21.1%) required hospitalization. By day 3 of hospitalization, 24% of patients were transferred to the intensive care unit, and around half of the remaining patients were discharged home. Ten patients died. Hospitalization risk was increased with older age, black race, male sex, former smoking history, diabetes, hypertension, chronic lung disease, poor socioeconomic status, shortness of breath, diarrhea, and certain medications (NSAIDs, immunosuppressive treatment). Hospitalization risk was reduced with prior flu vaccination. Model discrimination was excellent with an area under the curve of 0.900 (95% confidence interval of 0.886–0.914) in the development cohort, and 0.813 (0.786, 0.839) in the validation cohort. The scaled Brier score was 42.6% (95% CI 37.8%, 47.4%) in the development cohort and 25.6% (19.9%, 31.3%) in the validation cohort. Calibration was very good. The online risk calculator is freely available and found at https://riskcalc.org/COVID19Hospitalization/.

## Limitation

Retrospective cohort design.

## Conclusion

Our study crystallizes published risk factors of COVID-19 progression, but also provides new data on the role of social influencers of health, race, and influenza vaccination. In a context of a pandemic and limited healthcare resources, individualized outcome prediction through this nomogram or online risk calculator can facilitate complex medical decision-making.

## Introduction

Based on the latest estimates from the Centers for Disease Control (week ending in June 6, 2020), hospitalization rates in the United States due to Coronavirus disease of 2019 (COVID-19) range from 5.6/100,000 population in patients 4 years or younger and up to 273.8/100,000 population in those 65 years or older, posing a significant capacity challenge to the healthcare system. Strategies to address this challenge have focused on imposing social distancing to reduce viral transmission and increasing hospital bed capacity by drastically reducing usual occupancy, eliminating elective surgical procedures, and creating makeshift surge hospitals [1]. Social distancing practices have indeed helped in curbing the acute need for hospital beds– at least momentarily- but the long-term healthcare capacity requirements remain unclear as strategies for lifting restrictions and resuming normal activities are in flux. Improving our understanding of the clinical outcomes of patients infected with COVID-19 is therefore paramount. In addition, we need predictive algorithms that identify the COVID-19 patients at highest risk of progressing to severe disease to develop alternative approaches to safely manage

them. These predictive algorithms could also be used at a population level to guide social distancing and other risk limiting strategies in a focused fashion, rather than the blanket approaches of shelter-in-place for society.

Older age [2–3], smoking [4], and medical co-morbidities such as diabetes, hypertension, cardiovascular disease, chronic kidney disease, chronic lung disease [5], and cancer [5–6] have been correlated with disease worsening in patients who are *already* hospitalized with COVID-19. It is unclear how these comorbidities, or other patient characteristics, factor into clinical worsening that *leads to* hospitalization. Translating their significance at an individual patient care level when faced with a decision to hospitalize patients presenting with symptoms of COVID-19 is even more elusive. The end result is patients being told to go home from the emergency room only to return much more ill and be admitted days later, or patients hospitalized for observation for several days without any significant clinical deterioration.

We present the clinical characteristics and outcomes of patients with COVID-19, including a subset who were hospitalized. We also develop and validate a statistical model that can assist with individualized prediction of hospitalization risk for a patient with COVID-19. This model allows us to generate a visual statistical tool (a nomogram) that can consider numerous variables to predict an outcome of interest for an individual patient [7].

## Methods

### Patient selection

We included all patients, regardless of age, who had positive COVID-19 testing at Cleveland Clinic between March 8, 2020 and June 5, 2020. The study cohort included all Covid positive patients, whether they were hospitalized or not, from across the Cleveland clinic health system which includes >220 outpatient locations and18 hospitals in Ohio and Florida. As testing demand increased, we adapted our organizational policies and protocols to reconcile demand with patient and caregiver safety. Prior to March 18, any primary care physician could order a COVID-19 test. After that date, testing resources were streamlined through a "COVID-19 Hotline" which followed recommendations from the Centers for Disease Control (recommending to focus on high risk patients as defined by any of the following: Age older than 60 years old or less than 36 months old; on immune therapy; having comorbidities of cancer, end-stage renal disease, diabetes, hypertension, coronary artery disease, heart failure with reduced ejection fraction, lung disease, HIV/AIDS, solid organ transplant; contact with known COVID 19 patients; physician discretion was still allowed).

### Cleveland clinic COVID-19 registry

Demographics, co-morbidities, travel and COVID-19 exposure history, medications, presenting symptoms, socioeconomic measures, treatment, disease progression, and outcomes were collected. Registry variables were chosen to reflect available literature on COVID-19 disease characterization, progression, and proposed treatments, including medications thought to have benefits through drug-repurposing studies [8]. Capture of detailed research data was facilitated by the creation of standardized clinical templates that were implemented across the healthcare system as patients were seeking care for COVID-19-related concerns. Outcome capture was facilitated by a home monitoring program whereby patients who tested positive were called daily for 14 days after–test result to monitor their disease progression.

Data were extracted via previously validated automated feeds [9] from our electronic health record (Epic, Epic Systems Corporation) and manually by a study team trained on uniform sources for the study variables. The COVID-19 Research Registry team includes a "Reviewer"

group and a "Quality Assurance" group. The reviewers were responsible for manually abstracting and entering a subset of variables (signs and symptoms upon presentation) that cannot be automatically extracted from the electronic health record, and for verifying high-priority variables (co-morbidities) that have been automatically pulled into the database from the electronic health record. The Quality Assurance group provided an independent second layer of review. Study data were collected and managed using REDCap electronic data capture tools hosted at Cleveland Clinic [10–11]. REDCap (Research Electronic Data Capture) is a secure, web-based software platform designed to support data capture for research studies, providing 1) an intuitive interface for validated data capture; 2) audit trails for tracking data manipulation and export procedures; 3) automated export procedures for seamless data downloads to common statistical packages; and 4) procedures for data integration and interoperability with external sources.

This research was approved by the Cleveland Clinic Institutional Review Board (IRB# 20–283). Consent was waived by IRB.

## COVID-19 testing protocols

Nasopharyngeal and oropharyngeal swab specimens were both collected in all patients and pooled for testing by trained medical personnel. Given previous beliefs that co-infection with severe acute respiratory syndrome coronavirus 2 (SARS-CoV-2) and other respiratory viruses is rare [12–13], a reflex testing algorithm was implemented to conserve resources. All patient specimens were first tested for the presence of influenza A/B and respiratory syncytial virus (RSV), and only those negative for influenza and RSV were subsequently tested for SARS-CoV-2.

Infection with SARS-CoV-2 was confirmed by laboratory testing using the Centers for Disease Control and Prevention (CDC) reverse transcription polymerase chain reaction (RT-PCR) SARS-CoV-2 assay that was validated in the Cleveland Clinic Robert J. Tomsich Pathology and Laboratory Medicine Institute. This assay uses Roche Magnapure extraction and ABI 7500 DX PCR instruments. Between March 8 and 13, the tests were sent out to Lab-Corp, Burlington, North Carolina. All testing was authorized by the Food and Drug Administration under an Emergency Use Authorization (EUA), and in accordance with the guidelines established by the CDC.

## Statistical methods

Baseline data are presented as median [interquartile range [IQR]) and number (%)]. Continuous variables were compared using the Mann-Whitney U test, and categorical variables were compared using the Chi-square test. The outcome of interest was hospitalization anytime within three days of a positive COVID test. The model was built using a development cohort (patients with COVID positive test resulted before May 1, 2020), and subsequently tested in a validation cohort (patients with COVID positive test resulted between May 1 and June 5, 2020). This allowed us to test the model's validity over time. A full multivariable logistic model was initially constructed to predict hospital admission with COVID-19 based on demographic variables, comorbidities, immunization history, symptoms, travel history, lab variables, and medications that were identified pre-admission. For modeling purposes, methods of missing value imputation for labs variables were compared using median values and using values from multivariate imputation by chained equations (MICE) via the R package mice. Restricted cubic splines with 3 knots were applied to continuous variables to relax the linearity assumption. A least absolute shrinkage and selection operator (LASSO) logistic regression algorithm was performed to retain the most predictive features. A 10-fold cross validation method was

applied to find the regularization parameter lambda which gave the minimum mean cross-validated concordance index. Predictors with nonzero coefficients in the LASSO regression model were chosen for calculating predicted risk. The final model was internally validated by assessing the discrimination and calibration with 1000 bootstrap resamples. Discrimination was measured with the concordance index [14]. Calibration was assessed visually by plotting the nomogram predicted probabilities against the observed event proportions over a series of equally spaced values within the range of the predicted probabilities. The closer the calibration curve lies along the 45˚ line, the better the calibration. A scaled Brier score, called the index of predictive accuracy (IPA) [15], was also calculated, as this has some advantages over the more popular concordance index. The IPA ranges from -1 to 1, where a value of 0 indicates a useless model, and negative values imply a harmful model. We adhered to the TRIPOD checklist for reporting the prediction model [16].

We calculated sensitivity, specificity, positive addictive value, negative predictive value at different cutoffs of predicted risk. We used R, version 3.5.0 (R Project for Statistical Computing) [17], with tidyverse [18], mice [19], caret [20], and risk Regression [21] packages for all analyses. Statistical tests were 2-sided and used a significance threshold of P < .05. We included all COVID positive patients during the study period in this model development and validation to optimize model performance: no specific sample size calculations were performed.

### Sensitivity analyses

An outcome of "hospitalized versus not" allows us to predict the likelihood that the patient is actually getting admitted to the hospital. This decision, however, is influenced by multiple "non-medical" factors including bed availability, regulatory systems, and individual physician preferences. To test the applicability of our model towards a determination of whether a patient *should* have been admitted or not, we subdivided patients included in our validation cohort and development cohorts into 4 categories: A- hospitalized and not sent home within 24 hours; B- sent home (not initially hospitalized) but ultimately hospitalized within 1 week of being sent home; C- not hospitalized at all; D- hospitalized but sent home within 24 hours. In this construct, categories A and C represent patients who were "correctly managed", at categories B and D represent those who were "incorrectly managed". We then tested the discrimination of our model in each one of those categories separately.

No model recalibration was done.

## Results

### Patient characteristics and outcomes

4,536 patients tested positive during the study period, including 2,852 patient in the development cohort (DC) of whom 582 (20.4%) were hospitalized, and 1,684 patients in the validation cohort (VC) of whom 376 (22.3%) were hospitalized. Table 1 provides demographic, exposure, clinical, laboratory, social characteristics, and medication history of COVID-19 patients who were hospitalized versus those who completed their treatment on an outpatient basis in both the DC and VC. At the time of hospital admission, 260 patients were known to have COVID-19, while the results of the (RT-PCR) SARS-CoV-2 nasopharyngeal assay were still pending on 698. Six hundred and sixty five were admitted from the emergency room, 32 were transferred from other hospitals, and 261 were directly admitted from the outpatient areas. Overall outcomes illustrated in Fig 1 show the cumulative incidence of hospital discharge, transfer to intensive care unit, and death in our hospitalized cohort.

**Table 1. Detailed descriptive statistics of demographic, exposure, clinical, laboratory, social characteristics, and medication history of COVID-19 positive patients who were hospitalized versus not.** Statistically significant variables (p-value<0.05) are bolded. The development data is before 05/01 and the validation data is between 05/01 and 06/05. The percentages presented are per row.

| | Development Cohort | | | Validation Cohort | | |
|---|---|---|---|---|---|---|
| | Not hospitalized | hospitalized | p-value | Not hospitalized | hospitalized | p-value |
| **N** | **2270** | **582** | | **1308** | **376** | |
| **Demographics:** | | | | | | |
| Race (%) | | | **<0.001** | | | **<0.001** |
| Asian | 27 (77.1) | 8 (22.9) | | 8 (53.3) | 7 (46.7) | |
| Black | 498 (70.0) | 213 (30.0) | | 422 (68.0) | 199 (32.0) | |
| Other | 362 (93.5) | 25 (6.5) | | 239 (90.5) | 25 (9.5) | |
| White | 1383 (80.5) | 336 (19.5) | | 639 (81.5) | 145 (18.5) | |
| Male (%) | 1049 (76.5) | 323 (23.5) | **<0.001** | 556 (75.3) | 182 (24.7) | **0.049** |
| Ethnicity (%) | | | **<0.001** | | | **<0.001** |
| Hispanic | 326 (93.9) | 21 (6.1) | | 99 (84.6) | 18 (15.4) | |
| Non-Hispanic | 1677 (75.2) | 553 (24.8) | | 925 (72.8) | 345 (27.2) | |
| Unknown | 267 (97.1) | 8 (2.9) | | 284 (95.6) | 13 (4.4) | |
| Smoking (%) | | | **<0.001** | | | **<0.001** |
| Current Smoker | 136 (78.6) | 37 (21.4) | | 111 (71.2) | 45 (28.8) | |
| Former Smoker | 642 (74.4) | 221 (25.6) | | 301 (70.2) | 128 (29.8) | |
| No | 1182 (79.6) | 302 (20.4) | | 613 (78.8) | 165 (21.2) | |
| Unknown | 310 (93.4) | 22 (6.6) | | 283 (88.2) | 38 (11.8) | |
| Age (median [IQR]) Missing: 0.5% | 50.57 [35.75, 64.40] | 64.37 [54.83, 76.58] | **<0.001** | 45.57 [30.49, 65.93] | 64.94 [52.45, 76.78] | **<0.001** |
| **Exposure history:** | | | | | | |
| Exposed to COVID-19? YES (%) | 1725 (81.6) | 390 (18.4) | **<0.001** | 732 (78.3) | 203 (21.7) | 0.535 |
| Family member with COVID-19? YES (%) | 1565 (80.3) | 383 (19.7) | 0.161 | 557 (75.0) | 186 (25.0) | **0.021** |
| **Presenting symptoms:** | | | | | | |
| Cough? Yes (%) | 1889 (79.8) | 478 (20.2) | 0.576 | 662 (77.3) | 194 (22.7) | 0.781 |
| Fever? Yes (%) | 1534 (79.2) | 403 (20.8) | 0.472 | 505 (77.3) | 148 (22.7) | 0.838 |
| Fatigue? Yes (%) | 1479 (76.8) | 446 (23.2) | **<0.001** | 531 (73.9) | 188 (26.1) | **0.001** |
| Sputum production? Yes (%) | 1042 (78.8) | 280 (21.2) | 0.365 | 458 (75.5) | 149 (24.5) | 0.114 |
| Flu-like symptoms? Yes (%) | 1711 (80.0) | 429 (20.0) | 0.439 | 659 (78.9) | 176 (21.1) | 0.245 |
| Shortness of breath? Yes (%) | 1098 (72.3) | 421 (27.7) | **<0.001** | 379 (66.5) | 191 (33.5) | **<0.001** |
| Diarrhea? Yes (%) | 995 (78.6) | 271 (21.4) | 0.256 | 370 (74.0) | 130 (26.0) | **0.022** |
| Loss of appetite? Yes (%) | 1222 (77.2) | 360 (22.8) | **0.001** | 464 (73.0) | 172 (27.0) | **<0.001** |
| Vomiting? Yes (%) | 711 (81.7) | 159 (18.3) | 0.069 | 282 (75.2) | 93 (24.8) | 0.217 |
| **Co-morbidities:** | | | | | | |
| BMI (median [IQR]) Missing: 52.7% | 29.27 [25.73, 33.98] | 30.30 [26.29, 35.46] | **0.03** | 30.05 [25.71, 35.18] | 29.02 [24.80, 34.95] | 0.15 |
| COPD/emphysema? Yes (%) | 102 (58.0) | 74 (42.0) | **<0.001** | 43 (50.6) | 42 (49.4) | **<0.001** |
| Asthma? Yes (%) | 264 (67.9) | 125 (32.1) | **<0.001** | 198 (75.6) | 64 (24.4) | 0.419 |
| Diabetes? Yes %) | 358 (60.3) | 236 (39.7) | **<0.001** | 193 (57.1) | 145 (42.9) | **<0.001** |
| Hypertension? Yes (%) | 800 (65.6) | 419 (34.4) | **<0.001** | 462 (64.7) | 252 (35.3) | **<0.001** |
| Coronary artery disease? Yes (%) | 172 (57.3) | 128 (42.7) | **<0.001** | 125 (61.3) | 79 (38.7) | **<0.001** |
| Heart failure? Yes (%) | 122 (52.4) | 111 (47.6) | **<0.001** | 80 (51.6) | 75 (48.4) | **<0.001** |
| Cancer? Yes (%) | 230 (66.7) | 115 (33.3) | **<0.001** | 124 (72.1) | 48 (27.9) | 0.079 |
| Transplant history? Yes (%) | 10 (41.7) | 14 (58.3) | **<0.001** | 2 (22.2) | 7 (77.8) | **<0.001** |
| Multiple sclerosis? Yes (%) | 23 (76.7) | 7 (23.3) | 0.863 | 11 (68.8) | 5 (31.2) | 0.576 |
| Connective tissue disease? Yes (%) | 165 (69.3) | 73 (30.7) | **<0.001** | 47 (74.6) | 16 (25.4) | 0.658 |
| Inflammatory Bowel Disease? Yes (%) | 80 (72.1) | 31 (27.9) | 0.059 | 27 (75.0) | 9 (25.0) | 0.852 |
| Immunosuppressive disease? Yes (%) | 164 (59.0) | 114 (41.0) | **<0.001** | 125 (59.8) | 84 (40.2) | **<0.001** |

*(Continued)*

**Table 1.** (Continued)

| | Development Cohort | | | Validation Cohort | | |
|---|---|---|---|---|---|---|
| | Not hospitalized | hospitalized | p-value | Not hospitalized | hospitalized | p-value |
| N | 2270 | 582 | | 1308 | 376 | |
| **Vaccination history:** | | | | | | |
| Influenza vaccine? Yes (%) | 818 (72.5) | 311 (27.5) | <**0.001** | 423 (67.6) | 203 (32.4) | <**0.001** |
| Pneumococcal polysaccharide vaccine? Yes (%) | 264 (57.9) | 192 (42.1) | <**0.001** | 178 (56.3) | 138 (43.7) | <**0.001** |
| **Laboratory findings upon presentation:** | | | | | | |
| Pre-testing platelets (median [IQR]) Missing: 67.3% | 213.00 [163.00, 267.00] | 190.00 [153.25, 241.75] | <**0.001** | 213.00 [171.00, 270.50] | 207.00 [156.00, 273.00] | 0.266 |
| Pre- testing AST (median [IQR]) Missing: 72.0% | 28.00 [21.00, 40.00] | 36.00 [25.00, 52.00] | <**0.001** | 25.00 [20.00, 34.50] | 31.50 [22.00, 47.00] | <**0.001** |
| Pre- testing BUN (median [IQR]) Missing: 67.8% | 13.00 [10.00, 19.00] | 18.00 [12.00, 30.00] | <**0.001** | 13.00 [10.00, 18.00] | 19.00 [12.00, 30.75] | <**0.001** |
| Pre- testing Cholride (median [IQR]) Missing: 67.8% | 100.00 [98.00, 103.00] | 98.00 [95.00, 101.00] | <**0.001** | 101.00 [98.00, 103.00] | 99.00 [96.00, 103.00] | **0.004** |
| Pre- testing Creatinine (median [IQR]) Missing: 67.7% | 0.90 [0.74, 1.11] | 1.10 [0.84, 1.57] | <**0.001** | 0.87 [0.71, 1.11] | 1.05 [0.79, 1.48] | <**0.001** |
| Pre-testing hematocrit (median [IQR]) Missing: 67.4% | 40.60 [36.40, 44.12] | 40.00 [36.30, 43.80] | 0.421 | 39.35 [36.00, 42.50] | 39.00 [34.60, 42.70] | 0.285 |
| Pre- testing Potassium (median [IQR]) Missing: 67.2% | 4.00 [3.70, 4.20] | 4.00 [3.70, 4.40] | **0.034** | 3.90 [3.60, 4.20] | 4.00 [3.70, 4.40] | **0.005** |
| **Home medications:** | | | | | | |
| Immunosuppressive treatment? Yes (%) | 162 (67.8) | 77 (32.2) | <**0.001** | 69 (63.9) | 39 (36.1) | **0.001** |
| NSAIDS? Yes (%) | 388 (66.2) | 198 (33.8) | <**0.001** | 187 (61.1) | 119 (38.9) | <**0.001** |
| Steroids? Yes (%) | 192 (67.1) | 94 (32.9) | <**0.001** | 86 (63.7) | 49 (36.3) | <**0.001** |
| Carvedilol? Yes (%) | 38 (50.7) | 37 (49.3) | <**0.001** | 17 (53.1) | 15 (46.9) | **0.002** |
| ACE inhibitor? Yes (%) | 160 (62.5) | 96 (37.5) | <**0.001** | 83 (62.9) | 49 (37.1) | <**0.001** |
| ARB? Yes (%) | 128 (66.0) | 66 (34.0) | <**0.001** | 38 (53.5) | 33 (46.5) | <**0.001** |
| Melatonin? Yes (%) | 51 (60.0) | 34 (40.0) | <**0.001** | 24 (48.0) | 26 (52.0) | <**0.001** |
| **Social influencers of health:** | | | | | | |
| Population Per Sq Km* (median [IQR]) | 3.09 [2.68, 3.32] | 3.04 [2.67, 3.31] | 0.42 | 3.06 [2.60, 3.32] | 3.15 [2.77, 3.38] | <**0.001** |
| Median Income ($1000, median [IQR]) | 57.85 [44.78, 76.40] | 55.18 [36.27, 73.09] | <**0.001** | 50.80 [36.06, 65.76] | 41.67 [29.38, 64.08] | <**0.001** |
| Population Per Housing Unit (median [IQR]) | 2.29 [1.99, 2.61] | 2.15 [1.92, 2.40] | <**0.001** | 2.22 [1.93, 2.46] | 2.06 [1.79, 2.31] | <**0.001** |

* transformed as $\log_{10}(x+1)$

## Prediction modeling results

Imputation methods were evaluated with 1000 repeated bootstrapped samples. We found that models based on median imputation appeared to outperform those based on data from MICE imputation, so median imputation was selected for the basis of the final model. Variables that we examined and were not found to add value beyond those included in our final model for predicting hospitalization included exposure to COVID 19, other family members with COVID-19, fever, fatigue, sputum production, flu-like symptoms, recent international travel, coronary artery disease, heart failure, on immunosuppressive treatment, other heart disease, other lung disease, pneumovax vaccine, BUN, on angiotensin converting enzyme inhibitor, angiotensin receptor blocker, toremifene, and paroxetine. Model discrimination was excellent with an area under the curve of 0.900 (95% confidence interval of 0.886–0.914) in the development cohort, and 0.813 (0.786, 0.839) in the validation cohort. The scaled Brier score was 42.6% (95% CI 37.8%, 47.4%) in the development cohort and 25.6% (19.9%, 31.3%) in the validation cohort. The nomogram is presented in Fig 2, and an online version of the statistical

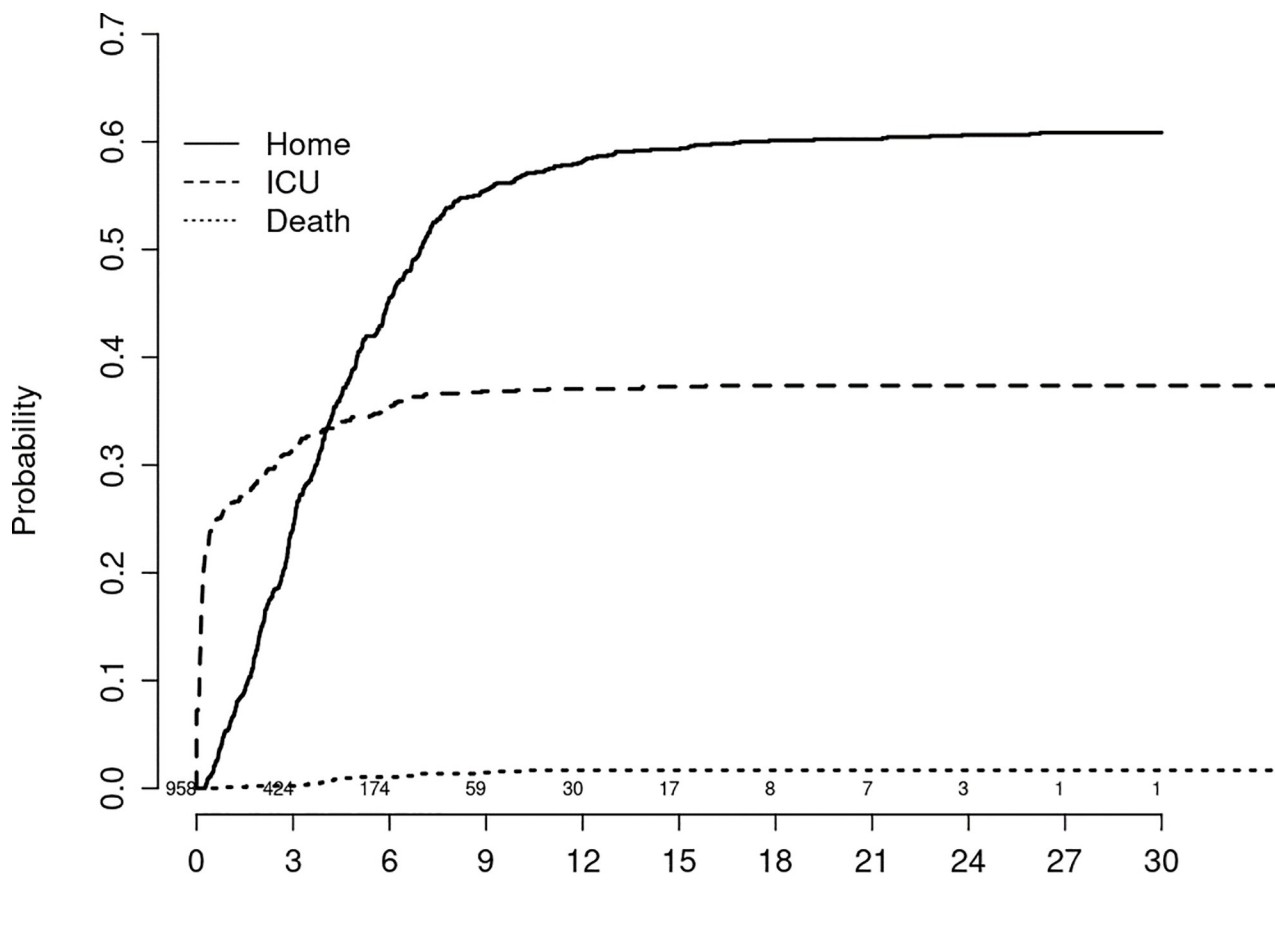

**Fig 1. This figure shows the cumulative incidence of each of the 3 outcomes (going home; transferred to ICU; death) following hospitalization in our COVID-19 cohort.** Values above the days from admission axis indicate numbers of patients at risk.

model (Fig 3) is available at https://riskcalc.org/COVID19Hospitalization/. The calibration curves are shown in Fig 4 and suggest that predicted risk matches observed proportions relatively well throughout the risk range. Table 2 shows the sensitivity, specificity, negative predictive value, and positive predictive value at different cutoffs of predicted risk.

## Sensitivity analysis

Appropriately managed patients represented the majority of the cohort: 750 patients were hospitalized with a length of stay that exceeded 24 hours (431 in DC and 319 in VC), and 3549 patients were not hospitalized at all (2258 in DC and 1291 in VC). A minority of patients (237 patients, 5.4%) fell in the category of inappropriate initial management: 208 had been initially sent home from the emergency room but were then admitted within 1 week of emergency room visit (151 in DC, 57 in VC), and 29 patients were hospitalized but then discharged within 24 hours (12 in DC, and 17 in VC). When tested in each one of those categories, the predictive model performed very well in the appropriately managed subgroup (area under the curve of 0.821), but its performance was inadequate in the 5.4% of patients who fell in the inappropriate initial management category.

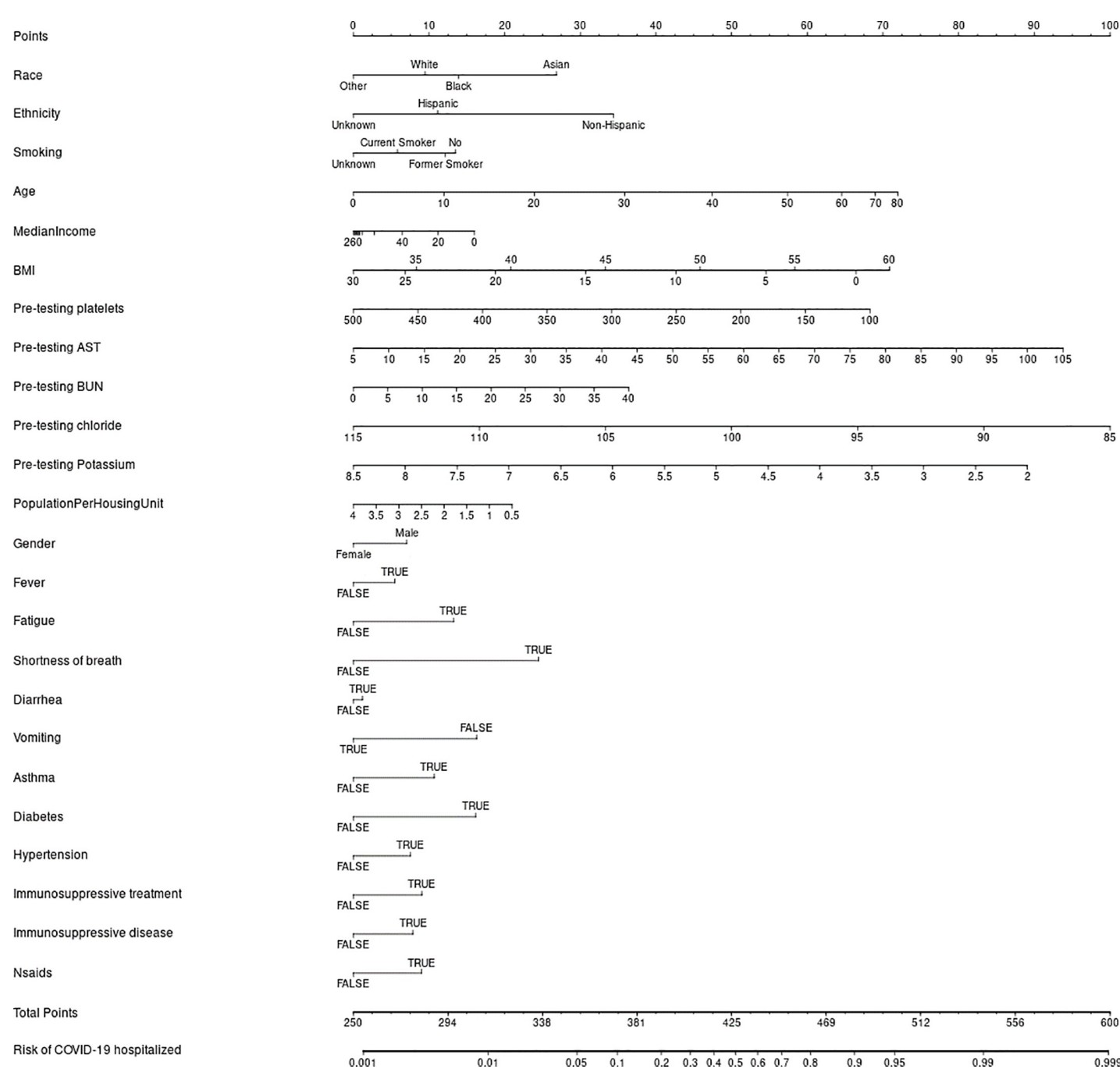

**Fig 2. A nomogram (graphical version of the model) is shown.** Line 1 is used to calculate the points that are associated with each of the predictor variables. Each subsequent line represents a predictor in the final model. The patient's characteristic is found on each line, and from it, a vertical line is drawn to find the points that are associated with each value. All the points are then totaled and located on second to last line. A vertical line is drawn down to the bottom line to locate the predicted risk of hospitalization produced by the model.

## Discussion

### Predictors of hospitalization

Our results confirm a higher risk of hospitalization with older age (median age in hospitalized patients of 65.5 years compared to 48.0 years in non-hospitalized patients), male sex (56.9% of

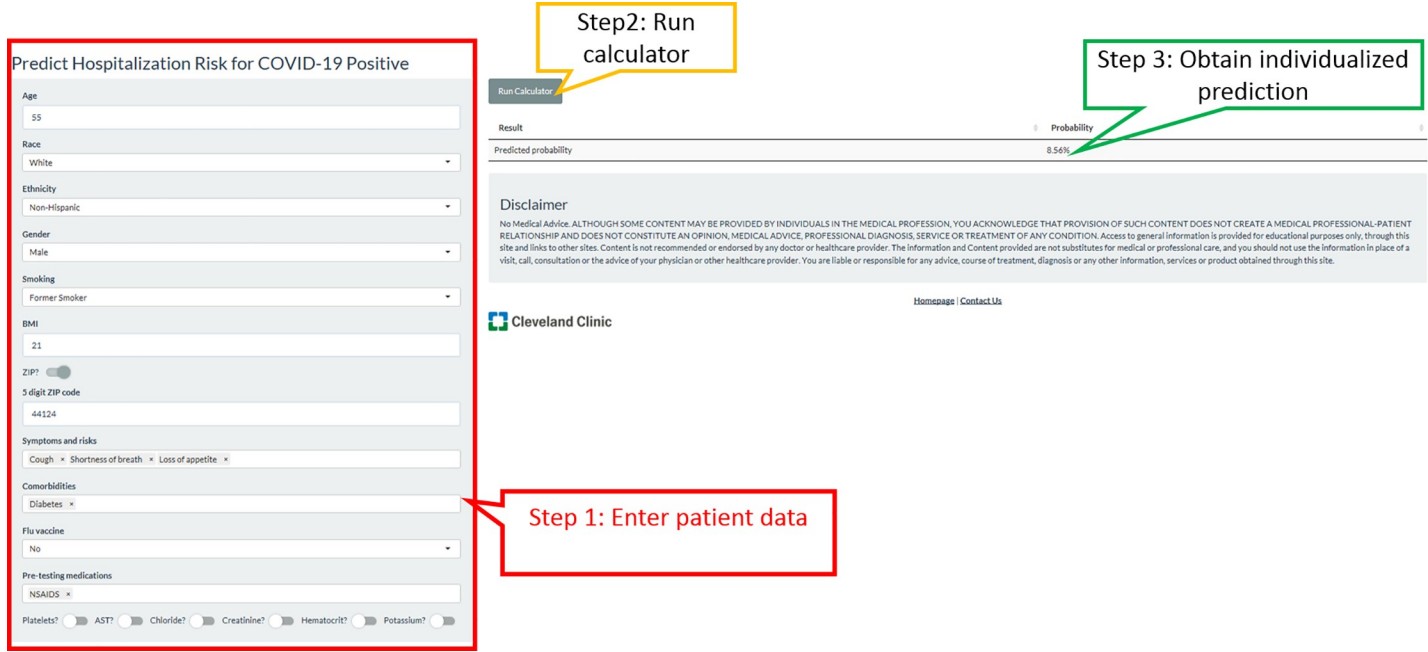

**Fig 3. Online risk calculator for risk of hospitalization from COVID-19, found at https://riskcalc.org/COVID19Hospitalization/.** The example here is a 55-year-old white male, former smoker, who presented with cough, shortness of breath, and loss of appetite. He has diabetes and received no vaccinations this year and is only on NSAIDs for some chronic joint pains. No labs are available yet. His predicted risk of hospitalization is 8.56%. If race is changed to Black, with all other variables remaining constant, his relative risk almost doubles to an absolute value of 17.22%.

hospitalized vs 48.3% of non-hospitalized), and medical co-morbidities most prominently hypertension, diabetes, and immunosuppressive disease (variables significant on univariable analysis in Table 1, but also relevant in final model). The significant association of shortness of breath and diarrhea with hospitalization may reflect the need for inpatient supportive care with these symptoms, regardless of the etiology. Beyond the expected, our results provide some insights that advance the existing literature:

1. _Smoking_: The World Health Organization warns of a higher morbidity for COVID-19 in smokers, and proposes multiple possible mechanisms including frequent touching of face and mouth during the act of smoking, sharing cigarettes, and underlying lung disease [22]. We found that former smokers rather than current smokers are at higher risk of COVID-related hospitalization (Table 1), favoring the underlying lung disease mechanism.

2. _Medications_: We found a higher risk of hospitalizations in COVID-19 patients who were on Angiotensin Converting Enzyme (ACE) inhibitors, or angiotensin II type-I receptor blockers (ARBs) on univariable analysis [16,23–24]. However, being on these medications did not influence the final multivariable model, suggesting that prior associations between ACEI's and ARBs with COVID severity may be confounded by the underlying medical co-morbidities (hypertension and diabetes) that are linked to highest COVID hospitalization rates, and which are most often treated with these same drugs. ACE2 can also be increased by thiazolidinediones and ibuprofen, potentially explaining the higher hospitalization risk seen in our patients on non-steroidal anti-inflammatory drugs (NSAIDs); in fact, the latest FDA guidance cautions against the use of NSAIDs in COVID patients [25]. Overall, we recommend caution using retrospective data to draw robust conclusions assigning causation to drugs vs underlying co-morbidity vs genetically driven ACE2 polymorphism. We

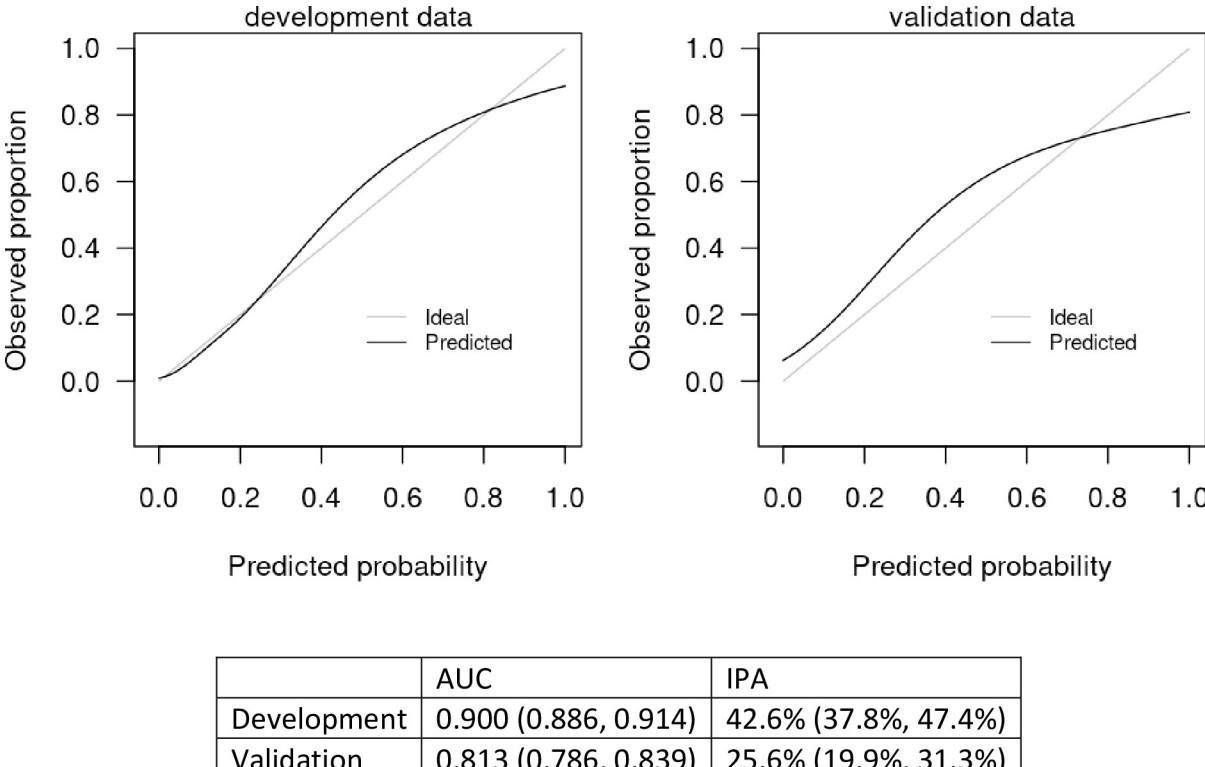

| | AUC | IPA |
|---|---|---|
| Development | 0.900 (0.886, 0.914) | 42.6% (37.8%, 47.4%) |
| Validation | 0.813 (0.786, 0.839) | 25.6% (19.9%, 31.3%) |

**Fig 4. Calibration curve for the model predicting likelihood of hospitalization.** The x-axis displays the predicted probabilities generated by the statistical model and the y-axis shows the fraction of the patients with COVID-19 who were hospitalized at the given predicted probability. The 45˚ line, therefore, indicates perfect calibration where, for example, at a predicted probability of 0.2 is associated with an actual observed proportion of 0.2. The solid black line indicates the model's relationship with the outcome. The closer the line is to the 45-degree line, the closer the model's predicted probability is to the actual proportion. As demonstrated, there is excellent correspondence between the predicted probability of a positive test and the observed frequency of hospitalization in COVID-19 (+) patients.

highlight the need for carefully designed, large observational studies or randomized clinical trials to address these critical questions.

3. _Race_: African American race was correlated with a higher hospitalization risk (36.2% of hospitalized vs 21% of non-hospitalized). This is consistent with a recent look at hospitalizations for COVID-19 across 14 states from March 1 to 30 [26]. Race data, which were available for 580 of 1,482 patients, revealed that African Americans accounted for 33 percent of the hospitalizations, but only 18 percent of the total population surveyed [26]. The authors proposed explanations like higher rates of medical co-morbidities, higher exposure risks, and distrust of the medical community as a postulated rationale. Our data, however, show

**Table 2. Sensitivity, specificity, positive predictive value, and negative predictive value of the model in the validation dataset at different cutoffs of predicted hospitalization risk.**

| | Sensitivity | Specificity | PPV | NPV |
|---|---|---|---|---|
| 10% | 0.769 | 0.726 | 0.447 | 0.916 |
| 30% | 0.519 | 0.918 | 0.646 | 0.896 |
| 50% | 0.388 | 0.963 | 0.749 | 0.846 |
| 70% | 0.253 | 0.979 | 0.772 | 0.820 |
| 90% | 0.117 | 0.992 | 0.800 | 0.796 |

that the effect of race on the individualized hospitalization risk prediction far outweighs that of any medical co-morbidity (Fig 2). It is already known that race influences the effectiveness of an immune response [27]. A deeper exploration of the underlying genetics and biology of race in the defense against and the response to a SARS-CoV-2 infection is needed. This should be paired with a deeper exploration of social influencers of health such as population per square kilometer, and population per household which were also relevant in our nomogram. In our online risk calculator, only the zip code entry is required: the relevant social influencers data are derived from the zip code by our program.

## Why do we need a prediction tool?

Given the multitude of risk factors discussed, the nomogram and online risk calculator assist with obviating challenges of translating complex information to patient-level clinical decision-making [28]. During a pandemic, with hospital beds in short supply, it is critical to empower front-line healthcare providers with tools that can supplement and support decision-making about who to admit. Advances in tele-health can be leveraged for home monitoring to guide care delivery in an outpatient setting for those determined to be low risk based on the nomogram calculation. Models like ours developed with data obtained through an automated abstraction from the electronic health record (EHR) offer the promise of integration within the EHR to facilitate rapid and efficient implementation into the clinical workflow. Such a strategy is a pragmatic application of overdue calls for a Learning Health System [29].

## How well does this nomogram perform?

Model performance, as measured by the concordance index, is excellent (c-statistic = 0.900). This level of discrimination is clearly superior to a coin toss or assuming all patients are at equivalent risk (both c-statistics = 0.5). The calibration of the model is excellent in both the DC and VC (see Fig 4). The metric that considers calibration, the IPA value, confirms that the model predicts substantially better than chance or no model at all. Overall, the model performs very well. Our next step will be to integrate this model into the clinical workflow.

## How can this model be integrated in a clinical workflow?

Manually abstracting data and inputting it in an online calculator is cumbersome in a busy clinical practice. Interpreting the prediction without some frame of reference is complex. However, failing to see beyond these hurdles risks wasting opportunities to innovate and improve patient care. It is therefore imperative to develop a clear implementation strategy that aligns with the existing clinical needs and clinical operations of a health organization. One could start by identifying the clinical problems that would benefit from this prediction tool, and reference the information in Table xx on sensitivity, specificity, positive predictive value, and negative predictive value at different prediction cutoffs to provide a framework for clinical application. An illustrative example now being explored from our own health system is the use of this calculator to tailor the intensity of home monitoring for COVID positive patients. Currently, every patient who tests positive for COVID is being called daily for 14 days to check on their symptoms and identify disease progression early enough for intervention. With only 20–30% of COVID positive patients progressing to the point of requiring hospitalization, the nurses can use our prediction tool to identify this high risk group and only call them daily, while reducing the intensity of follow-up with the rest.

## Limitations

This is not a multicenter study. It is important to note though that it includes all hospitals and outpatient facilities of the Cleveland Clinic Health System within the US (>220 outpatient locations and18 hospitals in Ohio and Florida) creating robust sampling of the COVID-19 population. As with any other statistical model, other hospital systems may elect to validate this model internally for their specific patient populations as they contemplate options for integrating it in their workflow. Given the alternative of no or constantly changing practice guidelines, implementation of this nomogram into our clinical workflow will allow prospective evaluation of its impact on patient care and outcomes. Our model includes age as a predictor: this may mitigate our ability to identify risk factors for disease progression specific in the younger population, and may underestimate the risks in the younger population with less severe disease and less likely to seek medical care. Lastly, although our model performs very well in the majority of COVID positive patients, more research is needed to optimize it for the sub group (5.4% of the total cohort in our series) with either delayed or unnecessary admission.

## Conclusions

Drivers of disease progression and worsening in COVID-19 are multiple and complex. We developed a statistical model with excellent predictive performance (c-statistic of 0.926) to individualize the hospitalization risk assessment at the patient level. This could help guide clinical decision-making and resource allocation.

## Supporting information

**S1 Checklist.**
(PDF)

## Author Contributions

**Conceptualization:** Lara Jehi, Steve Gordon, Michael W. Kattan.

**Data curation:** Alex Milinovich.

**Formal analysis:** Xinge Ji, Michael W. Kattan.

**Funding acquisition:** Lara Jehi.

**Investigation:** Lara Jehi.

**Methodology:** Lara Jehi, Steve Gordon, Michael W. Kattan.

**Project administration:** Lara Jehi.

**Resources:** Lara Jehi, Amy Merlino, James B. Young, Michael W. Kattan.

**Software:** Michael W. Kattan.

**Supervision:** Lara Jehi, James B. Young, Michael W. Kattan.

**Validation:** Michael W. Kattan.

**Visualization:** Michael W. Kattan.

**Writing – original draft:** Lara Jehi.

**Writing – review & editing:** Serpil Erzurum, Amy Merlino, Steve Gordon, James B. Young, Michael W. Kattan.

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
