## [Decision Letter · Decision Letter 0]

2 Jun 2020

PONE-D-20-11909

Characteristics, outcomes, and individualized prediction of hospitalization risk in 818 patients with COVID-19.

PLOS ONE

Dear Dr. Jehi,

Thank you for submitting your manuscript to PLOS ONE. After careful consideration, we feel that it has merit but does not fully meet PLOS ONE’s publication criteria as it currently stands. Therefore, we invite you to submit a revised version of the manuscript that addresses the points raised by the reviewers in their comments.

In my opinion, a crucial problem is that the risk calculator has been designed to identify patients that will be hospitalized, but not patients that require to be admitted. As the authors clearly explain in the manuscript, there are patients discharged home from emergency rooms that return more ill and need to be readmitted days later, or patients hospitalized for observation and sent home within three days with no intervention beyond supportive care. I wonder if the risk calculator could be useful to discriminate prospectively patients who need admission for treatment from those who can be safely managed as outpatients. Have the authors analyzed their model categorizing the patients in such subgroups?

In addition, the authors should revise the criteria of PLOS ONE for publication. Please, complete the TRIPOD checklist and include it in your submission.

We look forward to receiving your revised manuscript.

Kind regards,

Juan F. Orueta, MD, PhD

Academic Editor

PLOS ONE

Journal Requirements:

3. For studies involving humans categorized by race/ethnicity, age, disease/disabilities, religion, sex/gender, sexual orientation, or other socially constructed groupings, authors should:

a) Explicitly describe their methods of categorizing human populations,

b) Define categories in as much detail as the study protocol allows,

c) Justify their choices of definitions and categories,

d) Explain whether (and if so, how) they controlled for confounding variables such as socioeconomic status, nutrition, environmental exposures, or similar factors in their analysis.

Reviewers' comments:

Reviewer's Responses to Questions

**Comments to the Author**

1. Is the manuscript technically sound, and do the data support the conclusions?

Reviewer #1: Yes

Reviewer #2: Yes

Reviewer #3: Yes

Reviewer #4: Yes

2. Has the statistical analysis been performed appropriately and rigorously? 

Reviewer #1: Yes

Reviewer #2: Yes

Reviewer #3: Yes

Reviewer #4: Yes

3. Have the authors made all data underlying the findings in their manuscript fully available?

Reviewer #1: No

Reviewer #2: Yes

Reviewer #3: Yes

Reviewer #4: No

4. Is the manuscript presented in an intelligible fashion and written in standard English?

Reviewer #1: Yes

Reviewer #2: Yes

Reviewer #3: Yes

Reviewer #4: Yes

5. Review Comments to the Author

Reviewer #1: hank you for allowing me the opportunity to review this paper. This paper has led to the development of an online risk calculator for covid-19, which is very important in the current environment. Although I feel the paper has merit, some changes need to be made to improve the paper.

Author list:

- Please provide the affiliation details or author Merlino (Chief Medical Officer is not an affiliation)

Introduction:

Line 75  Clarify that this is in the United States, as the paper will be read by international readers

Line 101/102  Please remove the number of included patients. This should be left for the results section.

Lines 103-106  Belong in the methods sections, as you describe how the model was developed.

Methods:

Patient selection  Please describe how these patients were recruited, in which departments were they tested? Also add if this only included patients that were admitted or also patients who had positive tests but were not admitted

Line 134  Some studies suggest that there is a difference in sensitivity and specificity between the nasopharyngeeal and oropharyngeal swab. Did all patients have both swabs taken? If so, clarify that both speciemns were collected in all patients.

Line 170  Add a reference for the TRIPOD checklist

Results:

General comment: Numbers below 10 should be written as words, everything else in numerals.

Line 182 and 183: Is it known how many patients went home, fell ill again, but visited a different clinic to be admitted? Are you confident that all patients that fell ill came to the clinics included in your study? This is important to provide a accurate depiction.

Discussion:

Line 216: Does this also hold truth for current smokers with COPD in your dataset?

Line 274: Should other hospitals also include this model then in their clinical workflow or should they wait for it to be validated? Please add some info on this in the limitation section

Reviewer #2: The manuscript by Jehi L. et al. is focused on a crucial point in the decision making concerning patients affected by COVID 19. The ability to correctly stratify the risks could allow the physicians to perform a correct rule in/rule out or whenever patients need to be charged in the hospital to define the best care setting for each patient.

The manuscript is clearly written, data are well collected and the conclusions are well supported by novel and interesting results.

I think that the model proposed by the authors could be very useful. My concern regarding the model and the consequent risk calculator proposed relies in the relative complexity that for some care settings could be a problem.

Another methodological concern relies on the decision to test for SARS-CoV-2 only patients negative for influenza due, according the authors, to an anecdotes belief that coinfection is rare. However, to my knowledge, no scientific report has supported this belief. If the authors can support this concept with a reference should add it, otherwise they should test all suspected patients independently by the existence of a influenza A/B coinfection for avoiding possible statistical bias.

Also the idea to insert "age" in the model could be problematic. That's' not a lab value ... and speaks to the biases in the population cohort.. meaning that in a children's hospital, or any ED with a younger population (perhaps a suburban ED).. the scores would automatically reflect a lower risk for covid simply as a function of a younger aged population. The disease does not selectively choose its patients based on age.... although it does take a more aggressive course in older pts (who then are more likely to come to the ED). The net effect would be that the false negatives would intuitively be highest in the younger population.. which is not good. May be the authors should discuss this issue.

Concerning the data management rigourous statistical methodologies were followed for the analysis which are described in depth in the method section. However a major drawback is the absence of external validation of the model, and since the patient sample is relatively large, authors should think of splitting the cohort in a training and validation subset.

Moreover, the authors state in the Conclusion that their model is able “to individualize this risk assessment at the patient level”. The only risk really assessed by the model is the hospitalization because, for example, no indication is given on the outcome of the patients. I think the authors should point out this issue in the Discussion Section.

Reviewer #3: Dear Editor,

Thank you for giving me the opportunity to review the article: ”Characteristics, outcomes, and individualized prediction of hospitalization risk in 818 patients with COVID-19” by Jehi et al.

The article describes characteristics of patients testing positive for SARS-CoV-2 and tries to find predictors for hospital admission. The authors have used several advanced statistical approaches and handle them with great skill. Nevertheless, I have several major concerns that must be addressed:

1. The authors describe that the aim is to identify factors that could help in selecting patients that should be admitted. As the authors state: “The end result is patients being told to go home from the emergency room only to return much more ill and be admitted days later, or patients hospitalized for observation for several days without any significant clinical deterioration”. However, the outcome analyzed in the article, as I understand it, is those patients that de facto were admitted. I think the correct outcome would be those patients that should have been admitted as the authors themselves imply in the statement above. Patients that actually are admitted of course depends on the attending physician and as the authors discuss there can be both over- and under-admittance.

The fraction of patients that actually are admitted also heavily depends on the location and regulatory system of the country and location (and even on available hospital beds for the moment). In some countries, all persons with positive SARS-CoV-2 are admitted and in some countries only those that really need oxygen or care 24/7 are admitted. Therefore, having admittance as an outcome will only reflect the judgment of the attending physician at that location. This could partly be overcome be analyzing patients that were sent home after admittance without requiring oxygen within 1-3 days and no deterioration and also accounting for patients that were not admitted but came back and were later admitted (in the same or any other hospital) within e.g. 7 days (or died at home). For both these patient categories the judgment to admit or not were “wrong” and the patients should the adjudicated to their “right” category and not their actual category which were done in this study.

I would preferably like to see such a sensitivity analysis to really be able to discuss actual risk factors. If not such an analysis is made known risk factors will likely bias the results since the attending physician will be more likely to admit patients with known risk factors from the literature even if these are not true risk factors for outpatients.

There are some clues to this in the discussion that some patients were sent home and then later returned but it is unclear to what category in the LASSO regression these patients were adjudicated (their “right” category, i.e. to be admitted, or the “wrong” category, i.e to be sent home which were done in the first place). The same is true for the opposite. The authors describe that 50% of the patients admitted not requiring treatment in the ICU, were sent home within 3 days. To what category were these patients adjudicated? Was it in the first place a correct medical judgment to admit them since they most likely did not need medical care 24/7?

2. To me, as a non-American, it is unclear if the cohort covers the whole population in the catchment area; both for patients with and without any insurance and if there might have been differences in the likelihood of being sampled for SARS-CoV-2 or admitted based on this.

3. If a patient were sent home and then deteriorated and came back or died at home or were seeking medical care again would such a patient always be recognized in the cohort or could that patient seek another caregiver and not be accounted for? Are all deaths (no matter where) within 30 days from inclusion among the 818 patients included accounted for?

4. The cohort is based on those that were sampled for SARS-CoV-2. What were the sampling criteria? In some countries persons are sampled as part of screening programs, whereas in other countries only patients that are admitted to the hospital are sampled and no patients that are planned to go home from the ED or seek the GP are sampled. This will of course bias the result in different ways and needs to be discussed and clarified.

Minor comments

1. The article presents in the result section in the abstract that 11,686 patients were tested and the abstract speaks about a large cohort. However, the data presented only represents (correctly) the data of the 818 patients positive for COVID-19. It could be made a little bit clearer that this is actually the case. The complications et cetera, as death or ICU, are only analyzed among those 232 that were admitted making the cohort even smaller.

2. In the method section reference to R and the used packages are missing.

3. To me it is unclear how the multiple imputation was handled with the LASSO regression. Was imputation made on each imputed data set or were any algorithms like MIRL (Multiple Imputation Random Lasso) used?

4. Some variables are log-transformed in table 1. How were these variables entered in the multiple imputation and in the LASSO regression?

5. Even if “modern” statistical measures such as IPA et cetera were used it would be nice to have the sensitivity, specificity and AUC for the score in the bootstrapped cohorts.

6. Ideally, but I understand this is of course much more work, it would be nice to evaluate the score in a different cohort, preferably in another setting/country to validate the findings. Even more so, since the outcome was admittance with all the problems outlined above. The authors try to address this by bootstrapping the cohorts but this does not fully address these potentially biasing factors.

Again, thank you for letting me review this article.

Reviewer #4: Lara et al. aimed to characterise a large cohort of patients hospitalised with COVID-19, their

outcomes, and develop a statistical model that allows individualised prediction of future

hospitalisation risk for a patient newly diagnosed with COVID-19.

Following are my detailed comments:

1. The model temporal/external validation is a necessary step for generalisation of any predictive model. One can do the temporal validation on the remaining data from the same retrospective cohort.

2. There is no guidance what cutoff might be used in practice. This will enhance the applicability of this equation in a clinical setting. It would be good to report sensitivity, specificity, PPV, and NPV at several thresholds to facilitate complex medical decision-making.

3. I appreciate that nomograms and online risk calculator are developed to provide individualised hospitalisation risk for a patient newly diagnosed with COVID-19. It would be good to report the risk equation (at least as a supplementary material) that will help future external validation of this equation.

4. I can see LASSO logistic regression has been used to retain most predictive features for hospitalisation risk, but there is no evidence reported that the parsimonious model performed better than the full model.

6. PLOS authors have the option to publish the peer review history of their article (what does this mean?). If published, this will include your full peer review and any attached files.

Reviewer #1: No

Reviewer #2: Yes: jacopo maria legramante

Reviewer #3: No

Reviewer #4: Yes: Muhammad Faisal

---

## [Author Response · Author response to Decision Letter 0]

23 Jun 2020

June 18, 2020

We thank the reviewers for their thorough and meticulous comments. Below are our point by point responses.

Best regards

Lara Jehi and Mike Kattan.

Reviewer #1: Thank you for allowing me the opportunity to review this paper. This paper has led to the development of an online risk calculator for covid-19, which is very important in the current environment. Although I feel the paper has merit, some changes need to be made to improve the paper.

Response: We thank reviewer#1 for the kind comments. In addition to the point by point responses below, we want to make the reviewer aware of one additional major change in this revision done to address concerns of reviewers #2-4: We expanded our sample size to now include 4,536 Covid positive patients, and we divided those patients into a development cohort and validation cohort. 

Author list:

- Please provide the affiliation details or author Merlino (Chief Medical Officer is not an affiliation)

Response: Affiliation is now provided.

Introduction:

Line 75  Clarify that this is in the United States, as the paper will be read by international readers. Response: Clarified. This is in the United States

Line 101/102  Please remove the number of included patients. This should be left for the results section. Response: We removed the number of included patients, as requested.

Lines 103-106  Belong in the methods sections, as you describe how the model was developed. Response: We removed lines 103-106 from the introduction and moved the information to methods, as recommended by this reviewer.

Methods:

Patient selection  Please describe how these patients were recruited, in which departments were they tested? Also add if this only included patients that were admitted or also patients who had positive tests but were not admitted. Response: We significantly expanded the patient selection paragraph in this revision, as per the reviewer’s suggestion. We added the following clarifying section: “The study cohort thus included all Covid positive patients, whether they were hospitalized or not. As testing demand increased, we adapted our organizational policies and protocols to reconcile demand with patient and caregiver safety. Prior to March 18, any primary care physician could order a Covid 19 test. After that date, testing resources were streamlined through a “COVID-19 Hotline” which followed recommendations from the Centers for Disease Control (recommending to focus on high risk patients as defined by any of the following: Age older than 60 years old or less than 36 months old; on immune therapy; having comorbidities of cancer, end-stage renal disease, diabetes, hypertension, coronary artery disease, heart failure with reduced ejection fraction, lung disease, HIV/AIDS, solid organ transplant; contact with known Covid 19 patients; physician discretion was still allowed).”

Line 134  Some studies suggest that there is a difference in sensitivity and specificity between the nasopharyngeeal and oropharyngeal swab. Did all patients have both swabs taken? If so, clarify that both speciemns were collected in all patients. Response: We added the qualifier “in all patients”

Line 170  Add a reference for the TRIPOD checklist. Response: We added the reference as per the reviewer’s request: Collins GS, Reitsma JB, Altman DG, Moons KG. Transparent Reporting of a multivariable prediction model for Individual Prognosis or Diagnosis (TRIPOD): the TRIPOD statement [published correction appears in Ann Intern Med. 2015 Apr 21;162(8):600]. Ann Intern Med. 2015;162(1):55‐63. doi:10.7326/M14-0697

Results:

General comment: Numbers below 10 should be written as words, everything else in numerals. Response: We apologize to the reviewer for this mis-step. We corrected the spelling of the numbers in this revision.

Line 182 and 183: Is it known how many patients went home, fell ill again, but visited a different clinic to be admitted? Are you confident that all patients that fell ill came to the clinics included in your study? This is important to provide a accurate depiction. Response: We thank the reviewer for highlighting this important issue. It crystallized for us the need to provide more details on where the patients were admitted from, and their ultimate disposition. 

All patients who test positive for COVID in our cohort are then followed in our home monitoring program (now described in lines 157-159 of Methods) and are called by our nursing staff daily for 14 days after their test result. Their clinical progression is therefore documented in detail in our medical records and we feel confident about capturing their outcomes. We now added the sentence to the methods section:“Outcome capture was facilitated by a home monitoring program whereby patients who tested positive were called daily for 14 days after–test result to monitor their disease progression.”

In addition, we added a sensitivity analysis to clarify the different paths that patients took to get hospitalized: This was added to the methods section:’ Sensitivity analyses: An outcome of “hospitalized versus not” allows us to predict the likelihood that the patient is actually getting admitted to the hospital. This decision, however, is influenced by multiple “non-medical” factors including bed availability, regulatory systems, and individual physician preferences. To test the applicability of our model towards a determination of whether a patient should have been admitted or not, we subdivided patients included in our validation cohort and development cohorts into 4 categories: A- hospitalized and not sent home within 24 hours; B- sent home (not initially hospitalized) but ultimately hospitalized within 1 week of being sent home; C- not hospitalized at all; D- hospitalized but sent home within 24 hours. In this construct, categories A and C represent patients who were “correctly managed”, at categories B and D represent those who were “incorrectly managed”. We then tested the discrimination of our model in each one of those categories separately.”

This was added to the results section:’ Sensitivity analysis: Appropriately managed patients represented the majority of the cohort: 750 patients were hospitalized with a length of stay that exceeded 24 hours (431 in DC and 319 in VC), and 3549 patients were not hospitalized at all (2258 in DC and 1291 in VC). A minority of patients (237 patients, 5.4%) fell in the category of inappropriate initial management: 208 had been initially sent home from the emergency room but were then admitted within 1 week of emergency room visit (151 in DC, 57 in VC), and 29 patients were hospitalized but then discharged within 24 hours (12 in DC, and 17 in VC). When tested in each one of those categories, the predictive model performed very well in the appropriately managed subgroup (area under the curve of 0.821), but its performance was inadequate in the 5.4% of patients who fell in the inappropriate initial management category.”

We hope that this additional level of details satisfies the reviewer’s questions.

Discussion

Line 216: Does this also hold true for current smokers with COPD in your dataset? Response: Yes. Each predictor that was included in the final model had independent predictive value regardless of the others. This means that smoking history was relevant, regardless of whether the patient had COPD or not.

Line 274: Should other hospitals also include this model then in their clinical workflow or should they wait for it to be validated? Please add some info on this in the limitation section. Response: We thank the reviewer for this important suggestion. We now added the following limitations section: “As with any other statistical model, other hospital systems may elect to validate this model internally for their specific patient populations as they contemplate options for integrating it in their workflow.”

Reviewer #2: The manuscript by Jehi L. et al. is focused on a crucial point in the decision making concerning patients affected by COVID 19. The ability to correctly stratify the risks could allow the physicians to perform a correct rule in/rule out or whenever patients need to be charged in the hospital to define the best care setting for each patient.

The manuscript is clearly written, data are well collected and the conclusions are well supported by novel and interesting results.

Response: We thank the reviewer for the kind and encouraging comments.

I think that the model proposed by the authors could be very useful. My concern regarding the model and the consequent risk calculator proposed relies in the relative complexity that for some care settings could be a problem. Response: We thank the reviewer for highlighting these important points. We now add a section called “How can this model be integrated in a clinical workflow?” to the discussion. We start it by highlighting the limitation due to complexity, but then go on to describe how to implement the calculator with an illustrative example.

“How can this model be integrated in a clinical workflow? Manually abstracting data and inputting it in an online calculator is cumbersome in a busy clinical practice. Interpreting the prediction without some frame of reference is complex. However, failing to see beyond these hurdles risks wasting opportunities to innovate and improve patient care. It is therefore imperative to develop a clear implementation strategy that aligns with the existing clinical needs and clinical operations of a health organization. One could start by identifying the clinical problems that would benefit from this prediction tool, and reference the information in Table 2 on sensitivity, specificity, positive predictive value, and negative predictive value at different prediction cutoffs to provide a framework for clinical application. An illustrative example now being explored from our own health system is the use of this calculator to tailor the intensity of home monitoring for COVID positive patients. Currently, every patient who tests positive for COVID is being called daily for 14 days to check on their symptoms and identify disease progression early enough for intervention. With only 20-30% of COVID positive patients progressing to the point of requiring hospitalization, the nurses can use our prediction tool to identify this high risk group and only call them daily, while reducing the intensity of follow-up with the rest.”

Another methodological concern relies on the decision to test for SARS-CoV-2 only patients negative for influenza due, according the authors, to an anecdotes belief that coinfection is rare. However, to my knowledge, no scientific report has supported this belief. If the authors can support this concept with a reference should add it, otherwise they should test all suspected patients independently by the existence of a influenza A/B coinfection for avoiding possible statistical bias. Response: We now add two references to support the low rates of coinfection:

Schuchat A; CDC COVID-19 Response Team. Public Health Response to the Initiation and Spread of Pandemic COVID-19 in the United States, February 24-April 21, 2020. MMWR Morb Mortal Wkly Rep. 2020;69(18):551‐556. Published 2020 May 8. doi:10.15585/mmwr.mm6918e2 

Zwald ML, Lin W, Sondermeyer Cooksey GL, et al. Rapid Sentinel Surveillance for COVID-19 - Santa Clara County, California, March 2020. MMWR Morb Mortal Wkly Rep. 2020;69(14):419‐421. Published 2020 Apr 10. doi:10.15585/mmwr.mm6914e3

Also the idea to insert "age" in the model could be problematic. That's' not a lab value ... and speaks to the biases in the population cohort.. meaning that in a children's hospital, or any ED with a younger population (perhaps a suburban ED).. the scores would automatically reflect a lower risk for covid simply as a function of a younger aged population. The disease does not selectively choose its patients based on age.... although it does take a more aggressive course in older pts (who then are more likely to come to the ED). The net effect would be that the false negatives would intuitively be highest in the younger population.. which is not good. May be the authors should discuss this issue. Response: We thank the reviewer for raising this important question. It is indeed a delicate point because the risk for disease progression does increase with age as shown in our data and by others, so this increased risk needs to be reflected in a risk calculator aimed at the general population, but as the reviewer points out, the calculator would not then be specific to children and may underestimate aspects of disease severity that pertain to them. We now added this to the limitation section:

“Our model includes age as a predictor: this may mitigate our ability to identify risk factors for disease progression specific in the younger population, and may underestimate the risks in the younger population with less severe disease and less likely to seek medical care.”

Concerning the data management rigourous statistical methodologies were followed for the analysis which are described in depth in the method section. However a major drawback is the absence of external validation of the model, and since the patient sample is relatively large, authors should think of splitting the cohort in a training and validation subset. Response: We thank the reviewer for the suggestion which we feel has greatly improved the quality of our manuscript. In this revision, as suggested, we now include a validation cohort (new table 1). For the purposes of modeling performed in this paper, we divided the patients into a development cohort (COVID positive test resulted before May 1, 2020), and validation cohort (COVID positive test resulted between May 1 and June 5, 2020). As shown in our results, the model performed extremely well in both the validation and the development cohort’s: The area under the curve was 0.900 with a 95% confidence interval of (0.886, 0.914) in the development cohort, and 0.813 (0.786, 0.839) in the validation cohort. The IPA was similarly impressive with 42.6% (37.8%, 47.4%) in the development cohort and 25.6% (19.9%, 31.3%) in the validation cohort. We also show the calibration curves for both the development and validation cohorts. These additional findings increase our confidence in the stability and validity of our model’s performance over time.

Moreover, the authors state in the Conclusion that their model is able “to individualize this risk assessment at the patient level”. The only risk really assessed by the model is the hospitalization because, for example, no indication is given on the outcome of the patients. I think the authors should point out this issue in the Discussion Section. Response: We thank the reviewer for pointing this out. We now change in this claim to be more specific “to individualize hospitalization risk assessment at the patient level”.

Reviewer #3: Dear Editor,

Thank you for giving me the opportunity to review the article: ”Characteristics, outcomes, and individualized prediction of hospitalization risk in 818 patients with COVID-19” by Jehi et al.

The article describes characteristics of patients testing positive for SARS-CoV-2 and tries to find predictors for hospital admission. The authors have used several advanced statistical approaches and handle them with great skill. Nevertheless, I have several major concerns that must be addressed:

1. The authors describe that the aim is to identify factors that could help in selecting patients that should be admitted. As the authors state: “The end result is patients being told to go home from the emergency room only to return much more ill and be admitted days later, or patients hospitalized for observation for several days without any significant clinical deterioration”. However, the outcome analyzed in the article, as I understand it, is those patients that de facto were admitted. I think the correct outcome would be those patients that should have been admitted as the authors themselves imply in the statement above. Patients that actually are admitted of course depends on the attending physician and as the authors discuss there can be both over- and under-admittance.

The fraction of patients that actually are admitted also heavily depends on the location and regulatory system of the country and location (and even on available hospital beds for the moment). In some countries, all persons with positive SARS-CoV-2 are admitted and in some countries only those that really need oxygen or care 24/7 are admitted. Therefore, having admittance as an outcome will only reflect the judgment of the attending physician at that location. This could partly be overcome be analyzing patients that were sent home after admittance without requiring oxygen within 1-3 days and no deterioration and also accounting for patients that were not admitted but came back and were later admitted (in the same or any other hospital) within e.g. 7 days (or died at home). For both these patient categories the judgment to admit or not were “wrong” and the patients should the adjudicated to their “right” category and not their actual category which were done in this study.

I would preferably like to see such a sensitivity analysis to really be able to discuss actual risk factors. If not such an analysis is made known risk factors will likely bias the results since the attending physician will be more likely to admit patients with known risk factors from the literature even if these are not true risk factors for outpatients.

There are some clues to this in the discussion that some patients were sent home and then later returned but it is unclear to what category in the LASSO regression these patients were adjudicated (their “right” category, i.e. to be admitted, or the “wrong” category, i.e to be sent home which were done in the first place). The same is true for the opposite. The authors describe that 50% of the patients admitted not requiring treatment in the ICU, were sent home within 3 days. To what category were these patients adjudicated? Was it in the first place a correct medical judgment to admit them since they most likely did not need medical care 24/7? Response: We appreciate the fascinating comments made here by the reviewer, and fully agree with every point made. We considered different approaches to address these comments and hope that what we did below is satisfactory:

1- We significantly expanded our sample size in this revision (up from 818 COVID positive patients to 4,536 COVID positive) allowing us to create both a development and validation cohort for our model (New table 1), and to get more granularity on patient trajectories.

2- In addition, we added a sensitivity analysis to clarify the different paths that patients took to get hospitalized: This was added to the methods section:’ Sensitivity analyses: An outcome of “hospitalized versus not” allows us to predict the likelihood that the patient is actually getting admitted to the hospital. This decision, however, is influenced by multiple “non-medical” factors including bed availability, regulatory systems, and individual physician preferences. To test the applicability of our model towards a determination of whether a patient should have been admitted or not, we subdivided patients included in our validation cohort and development cohorts into 4 categories: A- hospitalized and not sent home within 24 hours; B- sent home (not initially hospitalized) but ultimately hospitalized within 1 week of being sent home; C- not hospitalized at all; D- hospitalized but sent home within 24 hours. In this construct, categories A and C represent patients who were “correctly managed”, at categories B and D represent those who were “incorrectly managed”. We then tested the discrimination of our model in each one of those categories separately.” 

3- This was added to the results section:’ Sensitivity analysis: Appropriately managed patients represented the majority of the cohort: 750 patients were hospitalized with a length of stay that exceeded 24 hours (431 in DC and 319 in VC), and 3549 patients were not hospitalized at all (2258 in DC and 1291 in VC). A minority of patients (237 patients, 5.4%) fell in the category of inappropriate initial management: 208 had been initially sent home from the emergency room but were then admitted within 1 week of emergency room visit (151 in DC, 57 in VC), and 29 patients were hospitalized but then discharged within 24 hours (12 in DC, and 17 in VC). When tested in each one of those categories, the predictive model performed very well in the appropriately managed subgroup (area under the curve of 0.821), but its performance was inadequate in the 5.4% of patients who fell in the inappropriate initial management category.”

4- We added the following to the limitations section “Lastly, although our model performs very well in the majority of COVID positive patients, more research is needed to optimize it for the sub group (5.4% of the total cohort in our series) with either delayed or unnecessary admission.” To highlight the challenges of accurate outcome prediction in the subgroup with delayed or unnecessary admission.

We hope that this additional level of detail satisfies the reviewer’s questions.

2. To me, as a non-American, it is unclear if the cohort covers the whole population in the catchment area; both for patients with and without any insurance and if there might have been differences in the likelihood of being sampled for SARS-CoV-2 or admitted based on this. Response: The reviewer raises a very important question. In contrast to countries with nationalized or centralized healthcare, no health system in the United States covers the whole population in a catchment area. On healthcare system (a nonprofit organization) captures though a sizable portion of this population: in Cleveland Ohio, and Weston Florida, our health system covers about 40-50% of the population; our patient demographics mirror those of the city population, and lack of health insurance does not prevent hospitalization. We therefore feel that our findings indeed reflect the biological drivers of disease progression requiring hospitalization.

3. If a patient were sent home and then deteriorated and came back or died at home or were seeking medical care again would such a patient always be recognized in the cohort or could that patient seek another caregiver and not be accounted for? Are all deaths (no matter where) within 30 days from inclusion among the 818 patients included accounted for? Response: All patients who test positive for COVID in our cohort are then followed in our home monitoring program (lines 157-159 of Methods) and are called by our nursing staff daily for 14 days after their test result. Their clinical progression is therefore documented in detail in our medical records and we feel confident about capturing their outcomes. We now added the sentence to the methods section:

“Outcome capture was facilitated by a home monitoring program whereby patients who tested positive were called daily for 14 days after–test result to monitor their disease progression.”

4. The cohort is based on those that were sampled for SARS-CoV-2. What were the sampling criteria? In some countries persons are sampled as part of screening programs, whereas in other countries only patients that are admitted to the hospital are sampled and no patients that are planned to go home from the ED or seek the GP are sampled. This will of course bias the result in different ways and needs to be discussed and clarified. Response: We appreciate the importance of this request. We added now in the methods section the paragraph below to clarify the criteria for testing in our institution: 

“The study cohort thus included all COVID positive patients, whether they were hospitalized or not. As testing demand increased, we adapted our organizational policies and protocols to reconcile demand with patient and caregiver safety. Prior to March 18, any primary care physician could order a Covid 19 test. After that date, testing resources were streamlined through a “COVID-19 Hotline” which followed recommendations from the Centers for Disease Control (recommending to focus on high risk patients as defined by any of the following: Age older than 60 years old or less than 36 months old; on immune therapy; having comorbidities of cancer, end-stage renal disease, diabetes, hypertension, coronary artery disease, heart failure with reduced ejection fraction, lung disease, HIV/AIDS, solid organ transplant; contact with known COVID 19 patients; physician discretion was still allowed).”

Minor comments

1. The article presents in the result section in the abstract that 11,686 patients were tested and the abstract speaks about a large cohort. However, the data presented only represents (correctly) the data of the 818 patients positive for COVID-19. It could be made a little bit clearer that this is actually the case. The complications et cetera, as death or ICU, are only analyzed among those 232 that were admitted making the cohort even smaller. Response: As suggested, we now deleted the number of patients who were tested, and only mention those of who were positive.

2. In the method section reference to R and the used packages are missing. Response: We added the reference as per the reviewer’s request: We used R, version 3.5.0 (R Project for Statistical Computing), with tidyverse, mice, caret, and riskRegression packages for all analyses. Statistical tests were 2-sided and used a significance threshold of P < .05.

3. To me it is unclear how the multiple imputation was handled with the LASSO regression. Was imputation made on each imputed data set or were any algorithms like MIRL (Multiple Imputation Random Lasso) used? Response: We created 10 imputed datasets using the mice package in R. All covariates, including those without missingness, were used in the imputation process. We conducted the LASSO regression on the 10 imputed datasets and calculated the average concordance index and compared with other imputation methods.

4. Some variables are log-transformed in table 1. How were these variables entered in the multiple imputation and in the LASSO regression? Response: We used the log-transformed values in the multiple imputation and in the LASSO regression.

5. Even if “modern” statistical measures such as IPA et cetera were used it would be nice to have the sensitivity, specificity and AUC for the score in the bootstrapped cohorts. Response: We thank the reviewer for this very useful suggestion. We now added table 2 with the sensitivity, specificity, PPV, and NPV at several thresholds, as requested.

6. Ideally, but I understand this is of course much more work, it would be nice to evaluate the score in a different cohort, preferably in another setting/country to validate the findings. Even more so, since the outcome was admittance with all the problems outlined above. The authors try to address this by bootstrapping the cohorts but this does not fully address these potentially biasing factors. Response: We thank the reviewer for the suggestion. In this revision, we now include a validation cohort (new table 1). For the purposes of modeling performed in this paper, we divided the patients into a development cohort (COVID positive test resulted before May 1, 2020), and validation cohort (COVID positive test resulted between May 1 and June 5, 2020). As shown in our results, the model performed extremely well in both the validation and the development cohort’s: The area under the curve was 0.900 with a 95% confidence interval of (0.886, 0.914) in the development cohort, and 0.813 (0.786, 0.839) in the validation cohort. The IPA was similarly impressive with 42.6% (37.8%, 47.4%) in the development cohort and 25.6% (19.9%, 31.3%) in the validation cohort. We also show the calibration curves for both the development and validation cohorts. These additional findings increase our confidence in the stability and validity of our model’s performance over time. We acknowledge that this is not the same as doing a validation in a cohort from another country, but this is the best that we could do within the scope of a revised submission.

Again, thank you for letting me review this article.

Reviewer #4: Lara et al. aimed to characterise a large cohort of patients hospitalised with COVID-19, their outcomes, and develop a statistical model that allows individualised prediction of future

hospitalisation risk for a patient newly diagnosed with COVID-19.

Following are my detailed comments:

1. The model temporal/external validation is a necessary step for generalisation of any predictive model. One can do the temporal validation on the remaining data from the same retrospective cohort. Response: We thank the reviewer for this important comment. We now include a validation cohort as detailed in our responses to prior reviewers and shown in a new table 1 and a new figure 4

2. There is no guidance what cutoff might be used in practice. This will enhance the applicability of this equation in a clinical setting. It would be good to report sensitivity, specificity, PPV, and NPV at several thresholds to facilitate complex medical decision-making. Response: We thank the reviewer for this very useful suggestion. We now added table 2 with the sensitivity, specificity, PPV, and NPV at several thresholds, as requested.

3. I appreciate that nomograms and online risk calculator are developed to provide individualised hospitalisation risk for a patient newly diagnosed with COVID-19. It would be good to report the risk equation (at least as a supplementary material) that will help future external validation of this equation. Response: We appreciate the interest but prefer not to publish also the formula. The formula is intellectual property and we would be open to collaborating with anyone who wants to validate. We have repeatedly shared our models with academic institutions, but avoiding public posting of our formula allows us to control commercial access.

4. I can see LASSO logistic regression has been used to retain most predictive features for hospitalisation risk, but there is no evidence reported that the parsimonious model performed better than the full model. Response: In the validation data, the parsimonious model had a higher AUC and IPA (0.813, 25.6%) compared with the full model (0.802, 23.7%).

---

## [Decision Letter · Decision Letter 1]

15 Jul 2020

PONE-D-20-11909R1

Characteristics, outcomes, and individualized prediction of hospitalization risk in 4,536 patients with COVID-19.

PLOS ONE

Dear Dr. Jehi,

Thank you for submitting your manuscript to PLOS ONE. After careful consideration, we feel that it has merit but does not fully meet all PLOS ONE’s publication criteria as it currently stands. Therefore, we invite you to submit a revised version of the manuscript as soon as possible.

The authors have been responsive to the comments raised by the reviewers. However, there is lack of a TRIPOD checklist form in the submission. In my opinion, this paper does not accomplish the whole set of TRIPOD recommendations. Please, observe the list of items, make any minor changes required in your manuscript and complete the form.

We look forward to receiving your revised manuscript.

Kind regards,

Juan F. Orueta, MD, PhD

Academic Editor

PLOS ONE

Reviewers' comments:

Reviewer's Responses to Questions

**Comments to the Author**

1. If the authors have adequately addressed your comments raised in a previous round of review and you feel that this manuscript is now acceptable for publication, you may indicate that here to bypass the “Comments to the Author” section, enter your conflict of interest statement in the “Confidential to Editor” section, and submit your "Accept" recommendation.

Reviewer #1: All comments have been addressed

Reviewer #2: All comments have been addressed

Reviewer #4: All comments have been addressed

2. Is the manuscript technically sound, and do the data support the conclusions?

Reviewer #1: Yes

Reviewer #2: Yes

Reviewer #4: Yes

3. Has the statistical analysis been performed appropriately and rigorously? 

Reviewer #1: Yes

Reviewer #2: Yes

Reviewer #4: Yes

4. Have the authors made all data underlying the findings in their manuscript fully available?

Reviewer #1: Yes

Reviewer #2: Yes

Reviewer #4: Yes

5. Is the manuscript presented in an intelligible fashion and written in standard English?

Reviewer #1: Yes

Reviewer #2: Yes

Reviewer #4: Yes

6. Review Comments to the Author

Reviewer #1: The authors have addressed my comments in a satisfactory manner. I have no further comments. Thank you

Reviewer #2: I would suggest only to the authors to consider a recent paper which could further support the need to correctly stratify the risks in patients possibly affected by COVID 19 (see Formica V et al. Complete blood count might help to identify subjects withhigh probability of testing positive to SARS-CoV-2. Clinical Medicine 2020; 20: 1-6, No 4 July 2020 DOI:10.7861/clinmed.2020-0373)

Reviewer #4: The authors have adequately addressed my comments and I recommend the manuscript for publication.

7. PLOS authors have the option to publish the peer review history of their article (what does this mean?). If published, this will include your full peer review and any attached files.

Reviewer #1: No

Reviewer #2: **Yes: **Jacopo M. Legramante

Reviewer #4: No

---

## [Author Response · Author response to Decision Letter 1]

16 Jul 2020

July 16, 2020

Response to reviewer:

PONE-D-20-11909R1

Characteristics, outcomes, and individualized prediction of hospitalization risk in 4,536 patients with COVID-19.

PLOS ONE

Dear Dr.Orueta,

Thank you for the favorable review of our revised manuscript. We submit here the requested minor revisions with:

1- your request to add the tripod checklist and make the corresponding minor revisions in the manuscript, and 

2- the only suggestion from the reviewer (reviewer #2) to add a new reference (now reference#28).

Looking forward to hopefully an expeditious final decision.

Best regards

Lara Jehi, MD, MHCDS and Mike Kattan, PhD

---

## [Editor Report · Decision Letter 2]

22 Jul 2020

PONE-D-20-11909R2

Characteristics, outcomes, and individualized prediction of hospitalization risk in 4,536 patients with COVID-19.

PLOS ONE

Dear Dr. Jehi,

Thank you for submitting your manuscript to PLOS ONE. After careful consideration, we feel that it has merit but there are still two minor points that must be addressed before publishing the manuscript. We invite you to submit a revised version as soon as possible.

Firstly, the title of the manuscript does not follow the TRIPOD recommendation. Please, mend it.

Besides, I have noticed that there are some mistakes in table 1 and the results section. According to the title of Table 1, *"the statistically significant variables (p value <0.05) are in bold."* It was true in the first version of the manuscript but not in the following ones. Also, I suppose there are several typos in the p-value columns since there are big differences in some variables (for example, diarrhea). Furthermore, the description on lines 257-8 *("The significant association of sputum production, shortness of breath and diarrhea with hospitalization ...")* corresponds to the p-values in the first draft but not in the second one.

We look forward to receiving your revised manuscript.

Kind regards,

Juan F. Orueta, MD, PhD

Academic Editor

PLOS ONE

---

## [Author Response · Author response to Decision Letter 2]

23 Jul 2020

July 23, 2020

Dear Dr.Orueta,

We are submitting here a revised manuscript based on your second editorial request for minor revisions.

1- We changed the title from “Characteristics, Outcomes, and individualized prediction of hospitalization risk in 4,536 patients with COVID-19” to “Development and validation of a model for individualized prediction of hospitalization risk in 4,536 patients with COVID-19”. Hopefully this satisfies your concern about aligning the title with TRIPOD.

2- We bolded the significant p-values in Table 1. The percentages in the table are presented by row. We added a clarifying statement to that effect in the legend. We verified that the p-values are all correct. We deleted “sputum production” from line 258.

Thank you for your consideration, and appreciate your review of these minor edits. 

Best

Lara Jehi and Mike Kattan.

---

## [Editor Report · Decision Letter 3]

28 Jul 2020

Development and validation of a model for individualized prediction of hospitalization risk in 4,536 patients with COVID-19.

PONE-D-20-11909R3

Dear Dr. Jehi,

We’re pleased to inform you that your manuscript has been judged scientifically suitable for publication and will be formally accepted for publication once it meets all outstanding technical requirements.

Kind regards,

Juan F. Orueta, MD, PhD

Academic Editor

PLOS ONE
---

## [Editor Report · Acceptance letter]

30 Jul 2020

PONE-D-20-11909R3 

Development and validation of a model for individualized prediction of hospitalization risk in 4,536 patients with COVID-19. 

Dear Dr. Jehi:

I'm pleased to inform you that your manuscript has been deemed suitable for publication in PLOS ONE. Congratulations! Your manuscript is now with our production department. 

Kind regards, 

on behalf of

Dr. Juan F. Orueta 

Academic Editor

PLOS ONE